# BRAF-Mutated Melanoma Cell Lines Develop Distinct Molecular Signatures After Prolonged Exposure to AZ628 or Dabrafenib: Potential Benefits of the Antiretroviral Treatments Cabotegravir or Doravirine on BRAF-Inhibitor-Resistant Cells

**DOI:** 10.3390/ijms252211939

**Published:** 2024-11-06

**Authors:** Valentina Zanrè, Francesco Bellinato, Alessia Cardile, Carlotta Passarini, Stefano Di Bella, Marta Menegazzi

**Affiliations:** 1Section of Biochemistry, Department of Neuroscience, Biomedicine and Movement Sciences, University of Verona, Strada Le Grazie 8, 37134 Verona, Italy; valentina.zanre@univr.it (V.Z.);; 2Section of Dermatology and Venereology, Department of Medicine, University of Verona, Piazzale Stefani 1, 37126 Verona, Italy; francesco.bellinato@univr.it; 3Clinical Department of Medical, Surgical and Health Sciences, University of Trieste, Piazzale Europa 1, 34127 Trieste, Italy; sdibella@units.it

**Keywords:** AXL, c-Met, BRAF, CRAF, SLC7A11, FTH1, GPX4, transferrin, retinoblastoma protein, cyclins

## Abstract

Melanoma is an aggressive cancer characterized by rapid growth, early metastasis, and poor prognosis, with resistance to current therapies being a significant issue. BRAF mutations drive uncontrolled cell division by activating the MAPK pathway. In this study, A375 and FO-1, BRAF-mutated melanoma cell lines, were treated for 4–5 months with RAF inhibitor dabrafenib or AZ628, leading to drug resistance over time. The resistant cells showed altered molecular signatures, with differences in cell cycle regulation and the propensity of cell death. Dabrafenib-resistant cells maintained high proliferative activity, while AZ628-resistant cells, especially A375 cells, exhibited slow-cycling, and a senescent-like phenotype with high susceptibility to ferroptosis, a form of cell death driven by iron. Antiretroviral drugs doravirine and cabotegravir, known for their effects on human endogenous retroviruses, were tested for their impact on these resistant melanoma cells. Both drugs reduced cell viability and colony formation in resistant cell lines. Doravirine was particularly effective in reactivating apoptosis and reducing cell growth in highly proliferative resistant cells by increasing tumor-suppressor proteins p16^Ink4a^ and p27^Kip1^. These findings suggest that antiretroviral drugs can influence apoptosis and cell proliferation in RAF-inhibitor-resistant melanoma cells, offering potential therapeutic strategies for overcoming drug resistance.

## 1. Introduction

Melanoma accounts for just 1% of all skin cancers, but it is highly aggressive and the leading cause of skin cancer-related deaths. Over the past 50 years, melanoma incidence in Europe has increased by 3–8% annually [1,2]. Approximately 50% of melanomas harbor harmful mutations in the BRAF kinase, which regulates the downstream mitogen-activated protein kinase (MAPK) signaling pathway [3]. RAF (V-raf murine sarcoma viral oncogene) proteins, downstream effectors of RAS (Rat sarcoma virus), are serine/threonine kinases that phosphorylate MEKs (MAP2Ks), which in turn activate ERKs (extracellular signal-regulated kinases). Under physiological conditions, the signaling pathway is transiently active. The MAPK activation is initiated extracellularly by the binding of growth factors to tyrosine kinase receptors, leading to the activation of RAF/MEK/ERK proteins [4], which generate signals that promote cell growth. In contrast, BRAF mutations result in a constitutively active MAPK signaling pathway that increases kinase activity up to 500-fold, ultimately driving cellular transformation [5,6]. The RAF kinase family consists of ARAF, BRAF, and CRAF, with BRAF being the primary activating kinase and the most frequently mutated [7].

Targeted therapy with BRAF inhibitors (BRAFi) has significantly improved response rates and extended median progression-free survival by several months compared to non-targeted chemotherapy. Second-generation BRAFi, designed as ATP-competitive small molecules, include vemurafenib and dabrafenib [8], which were approved by the Food and Drug Administration over a decade ago for treating patients with advanced BRAF-mutated metastatic melanoma. These BRAFi are predicted to be ineffective against dimeric RAF [9] because the inhibitor stabilizes the αC-helix in the inactive (OUT) position, a configuration that is not sterically permissible for both protomers in a RAF dimer. Consequently, when an αC-OUT inhibitor binds to the first protomer, it forces the αC-helix of the second protomer into the active (IN) position [10]. A third generation of targeted drugs has been developed to address the limitations of earlier therapies. AZ628, a member of this generation, is a selective and potent small molecule designed to target BRAF-mutated malignancies [11,12,13]. AZ628 stabilizes the αC-helix of the first protomer in the active (IN) position, allowing a second drug molecule to bind the second protomer, leading to the formation of imperfect dimers via a non-competitive allosteric inhibition mechanism [10]. As a result, αC-IN RAF inhibitors like AZ628 can hinder both BRAF and CRAF dimers, classifying them as ‘pan-RAF inhibitors,’ which may be particularly effective in RAF-mutant tumor cells where CRAF becomes the primary activator of ERK signaling [9,10]. However, increased CRAF protein levels can drive resistance to AZ628 [14]. To date, there are no ongoing trials studying the efficacy and safety of AZ628 in humans.

Despite initial benefits, the effectiveness of RAFi treatment is generally temporary, as patients eventually develop resistance, leading to disease progression [14,15]. Understanding the mechanisms of RAFi resistance is crucial for improving treatment outcomes in BRAF-mutated melanomas. Previous studies have explored acquired resistance to dabrafenib in melanoma cells and patients [16,17,18], and to AZ628 in cells [14]. Therefore, we studied and characterized the molecular features of two BRAF-mutated human cell lines, A375 and FO-1, following long-term treatment with increasing concentrations of dabrafenib or AZ628, and compared them to their respective parental cell populations. Our findings revealed significantly different molecular responses to each drug treatment.

Combination therapies targeting multiple pathways, such as DNA-damaging agents and BRAF inhibitors, have demonstrated enhanced tumor cell death both in vitro and in vivo, while also preventing melanoma re-growth after treatment discontinuation [19]. However, given the significant side effects of chemotherapy, there is an increasing preference to use more tolerable compounds, and combination treatments may lead to heightened side effects.

Recently, we investigated the potential benefits of administering antiretroviral drugs on the growth and invasive capabilities of various human melanoma cell lines, including A375 and FO-1 cells [20]. In this previous study, we demonstrated that lamivudine, doravirine, and cabotegravir affected the transcriptional activity of human endogenous retroviruses (HERV-K), leading to reduced cell growth, decreased clonogenic activity, and the induction of apoptosis or ferroptosis [20]. Although antiretroviral drugs have not yet been approved for melanoma treatment, several reverse transcriptase inhibitors have been shown to influence cell growth and differentiation in different cancers [21]. Remarkably, it is widely recognized that many antiretroviral drugs are safe for long-term use, as patients with HIV/AIDS typically require lifelong antiretroviral therapy, often with minimal side effects. Based on our previous findings, we selected the two most effective antiretroviral drugs to test their efficacy on A375 and FO-1 cells after they had developed resistance to dabrafenib or AZ628. Doravirine, a recently approved non-nucleoside reverse transcriptase inhibitor, and cabotegravir, a potent integrase strand transfer inhibitor, both retained anti-melanoma activity in melanoma cells resistant to targeted therapy. Their molecular effects on cell growth, and cell death pathways have been investigated.

## 2. Results

### 2.1. Cells’ Sensitivity Following Prolonged Treatment with RAF Inhibitors

To mimic in vivo melanoma-acquired drug resistance, A375 and FO-1 cells were treated with increasing concentrations of dabrafenib or AZ628 (range: 10 nM–5 µM). After 4–5 months of treatment, the cells were tested for their sensitivity to dabrafenib or AZ628 and used for further experiments. In all cases, two days before starting new experiments, treatment with RAF inhibitors was stopped by changing the cell culture medium.

### 2.2. Cell Viability Assay

Our initial aim was to investigate whether A375 and FO-1 cells remained sensitive after prolonged treatment with increasing concentrations of the RAF inhibitors dabrafenib or AZ628. All experiments were conducted 72 h after drug administration.

To assess the viability of A375 and FO-1 cells, we performed a DAPI (4′,6-diamidino-2-phenylindole dihydrochloride) fluorescence assay. This method provides higher accuracy by selectively staining nucleated cells, making it suitable for cell line samples with varying viability, including those with high debris levels [22].

As expected, both A375 and FO-1 parental cells (A375P and FO-1P) are highly sensitive to the cytotoxic effects of dabrafenib or AZ628. Treatment with dabrafenib reduced cell viability to 41.6% in A375 and 33.5% in FO-1 at 10 nM (S.D. ± 8.3 and 5.6, respectively). At 50 nM, the viability rate decreased further to 22.1% in A375 and 26.7% in FO-1 (S.D. ± 1.7 and 1.8). Similarly, AZ628 reduced cell viability to 48.8% in A375P and 46.7% in FO-1P at 10 nM (S.D. ± 5.1 and 3.7) and to 32.3% and 25.7%, respectively, at 50 nM (S.D. ± 4.8 and 2.3) compared to untreated controls (set at 100%) (Figure 1).

Conversely, administration of either dabrafenib or AZ628 at the same concentrations (10 and 50 nM) did not reduce cell viability in the cell lines treated for 4–5 months with increasing concentrations of each RAF inhibitor (Figure 2). Our results suggest that both cell lines have developed drug resistance. The resistant cell populations are designated as A375R and FO-1R for both inhibitors, with specific sub-designations as A375DR and FO-1DR (dabrafenib-resistant) or A375AR and FO-1AR (AZ628-resistant), in comparison to the parental A375P and FO-1P cells. Viability for A375DR was 103.5% at 10 nM and 98.6% at 50 nM (S.D. ± 0.8 and 1.1, respectively), while for FO-1DR, it was 94.6% at 10 nM and 99.7% at 50 nM (S.D. ± 2.3 and 3.7). Similarly, viability was 94.8% at 10 nM in A375AR and 94.5% at 50 nM (S.D. ± 2.3 and 6.4); in FO-1AR, it reached 99.6% at 10 nM and 78.7% at 50 nM (S.D. ± 4.1 and 19.1).

Next, we aim to test the effects of two different antiretroviral drugs on cell viability, specifically in the drug-resistant cell populations. Consistent with previous findings at lower concentrations [20], we confirmed that doravirine (10, 20 µM) and cabotegravir (6, 12 µM) significantly reduce cell viability in both parental cell lines. Notably, in A375P, cabotegravir lowered cell viability to 50.9% at 6 µM and 38.9% at 12 µM (S.D. ± 1.7 and 6.4, respectively). Doravirine reduced cell viability to 56.8% at 10 µM and 51.7% at 20 µM (S.D. ± 2.6 and 3.3). In FO-1P cells, cabotegravir diminished cell viability to 62.5% at 6 µM and 60.1% at 12 µM (S.D. ± 0.6 and 3.2), while doravirine decreased the survival rate to 81.7% at 10 µM and 80% at 20 µM (S.D. ± 2.6 and 6.0) (Figure 1). Interestingly, both antiretroviral drugs remain effective in resistant cell populations, with a particularly strong effect observed in A375DR cells, in which cabotegravir decreased cell survival to 40.9% at 6 µM and 40.5% at 12 µM (S.D. ± 5.1 and 6.2, respectively). Similarly, doravirine reduced cell survival to 56.9% at 10 µM and 40.5% at 20 µM (S.D. ± 4.4 and 5.4) (Figure 2). Although the viability of A375AR and FO-1R was also significantly affected, the effect was less pronounced. Specifically, cabotegravir reduced A375AR cell survival to 82.4% at 6 µM and 77.8% at 12 µM (S.D. ± 5.5 and 2.8, respectively), whereas doravirine decreased cell viability to 77% at 10 µM and 74.5% at 20 µM (S.D. ± 4.8 and 2.2). In FO-1DR cells, cabotegravir reduced viability to 66.6% at 6 µM and 63.7% at 12 µM (S.D. ± 7.5 and 8.4, respectively); meanwhile, doravirine reduced the survival rate to 71% at 10 µM and 63.7% at 20 µM (S.D. ± 8.2 and 5.0). In FO-1AR cells, cabotegravir reduced viability to 67.2% at 6 µM and 61.8% at 12 µM (S.D. ± 3.2 and 2.0), while doravirine reduced cell viability to 63.6% at 10 µM and 55.5% at 20 µM (S.D. ± 1.0 and 2.8) (Figure 2).

### 2.3. Administration of Antiretroviral Drugs and RAF Inhibitors Affect HERV-K Pol and Env Gene Expression

As we recently reported [20], doravirine and cabotegravir were shown to reduce the expression of human endogenous retrovirus-K (*HERV-K*) genes, *Pol* and *Env*, in A375P and FO-1P cell lines.

In the present work, we examine the effects of dabrafenib and AZ628 on *Pol* and *Env* gene expression in both parental and resistant cell lines.

As shown in Figure 3, real-time PCR assays demonstrate that treatment with both dabrafenib and AZ628 at a concentration of 50 nM resulted in a significant increase in *Pol* gene expression in A375P and FO-1P cells. However, the *Env* gene expression showed a noticeable, though not statistically significant, increase in the same parental cells (Figure 3, Top). Accordingly, the baseline expression of both *Pol* and *Env* genes increased following the development of RAFi resistance in the untreated control cells (Figure 3, Bottom). This suggests a potential involvement of *HERV-K* gene reactivation in the progression and worsening of the melanoma phenotype.

Moreover, in both A375DR and FO-1DR, a further increase in *Env* gene expression was observed after the administration of 50 nM dabrafenib (Figure 4). Meanwhile, in AZ628-resistant cell lines, a significant increase in both *Pol* and *Env* gene expression was detected after AZ628 treatment in FO-1AR cells, with no significant changes observed in A375AR cells (Figure 4).

Since RAF inhibitors appear to generally reactivate HERV-K gene expression in both parental and resistant cell populations, we used real-time PCR assays to evaluate whether the antiretroviral drugs doravirine and cabotegravir could reduce *HERV-K* gene expression levels in RAFi-resistant cell lines. Following treatment with 20 µM doravirine or 12 µM cabotegravir in A375R and FO-1R cell lines, a general reduction in *HERV-K* gene expression was observed across all four resistant cell populations, despite some variability (Figure 4). These findings suggest that the antiretroviral drugs remain effective even in the context of elevated *HERV-K* expression associated with RAF-inhibitor resistance.

### 2.4. Molecular Characterization of Melanoma Cells Resistant to Dabrafenib or AZ628

We first characterized the phenotypes of A375R and FO-1R by evaluating the expression levels of proteins involved in proliferative signaling, cell cycle, and cell death. These data were compared to those obtained from A375P and FO-1P cells, respectively. All the cell populations were treated for 72 h with the highest concentration previously tested on the parental cells (50 nM of either dabrafenib or AZ628).

In both parental cell lines, immunoblot analysis showed that treatment with dabrafenib or AZ628 significantly reduced the expression of BRAF1 and CRAF, as well as the effector kinases ERKs, both in their phosphorylated (activated) forms, and total protein levels (Figure 5). Additionally, two cell surface tyrosine kinase receptors constitutively expressed in melanoma cells, AXL (tyrosine-protein kinase receptor UFO) and c-Met (mesenchymal–epithelial transition factor) [23], were also downregulated by either dabrafenib or AZ628 in the parental cell populations (Figure 5).

Notably, in A375DR and FO-1DR cells, the expression levels of all these proteins returned to high levels, and further treatment with 50 nM dabrafenib was completely ineffective. These malignant features correlate with cell cycle reactivation in A375DR and FO-1DR cells, whether treated with dabrafenib or not. Indeed, the phosphorylation levels of retinoblastoma protein (pRb) and M-phase inducer phosphatase 3 (pCDC25C), as well as the total expression of cyclin D1 and cyclin A2, were reduced only in parental cells, but not in A375DR or FO-1DR cells, following dabrafenib treatment (Figure 6).

We also examined the expression levels of proteins involved in the regulation of cell death, focusing on apoptosis and ferroptosis. The expression of cleaved poly (ADP-ribose) polymerase (cPARP), a substrate cleaved by effector caspases 3 and 7, increases during apoptosis [24]. The mitochondrial carrier homolog 2 (MTCH2) protein, which facilitates the recruitment of pro-apoptotic proteins to the mitochondria, is essential for apoptosis induction, as deletion of the MTCH2 gene impairs this process [25]. In parental cells, cPARP levels increased following treatment with both RAF inhibitors, confirming their pro-apoptotic effect (Figure 7). In A375DR and FO-1DR cells, cPARP expression remained higher in both untreated or dabrafenib-treated conditions than in each corresponding parental untreated cell populations and was comparable to the levels observed in dabrafenib-treated parental cells (Figure 7). However, the elevated apoptosis levels in dabrafenib resistant cells were not explained by changes in the expression of anti-apoptotic proteins Bcl-2 (B-cell lymphoma 2) and Bcl-xL (B-cell lymphoma-extra-large), or the pro-apoptotic protein Bax (Bcl-2-like protein 4).

The propensity of A375P cells to undergo ferroptosis increased following treatment with dabrafenib, as indicated by elevated expression of transferrin and HO-1 (heme oxygenase-1), both of which contribute to augmented intracellular iron levels. This increase was associated with decreased levels of FTH1 (ferritin heavy chain 1) and reductions in GPX4 (glutathione peroxidase 4) and SLC7A11 (cystine-glutamate antiporter, also known as xCT). In contrast, A375DR cells exhibited a reduced likelihood of undergoing ferroptosis, as evidenced by lower transferrin levels and elevated SLC7A11 expression (Figure 7). Conversely, FO-1P cells treated with dabrafenib showed a lower propensity for ferroptosis, maintaining high levels of GPX4 and FTH1. Although FO-1DR cells displayed high levels of transferrin and HO-1, they appear protected from ferroptosis by elevated levels of SLC7A11, GPX4, and FTH1 (Figure 7).

A distinct phenotype was observed in A375AR cells. This population, resistant to the pan-Raf inhibitor AZ628, exhibited consistently low levels of signaling kinases (BRAF1, CRAF, ERKs, AXL, and c-Met), both in the presence and absence of the drug in the culture medium (Figure 5). Correspondently, the expression of cell cycle regulatory proteins (pRb, pCDC25C, cyclin D1, and cyclin A2) was similarly reduced, suggesting a very low proliferation rate (Figure 6).

While A375P cells treated with AZ628 showed increased cPARP expression, A375AR cells displayed nearly undetectable levels of cPARP, regardless of the presence of AZ628 in the culture medium. A375P cells treated with AZ628 were susceptible to cell death through ferroptosis, as evidenced by high transferrin expression and lower levels of SLC7A11 and FTH1 compared to untreated A375P cells (Figure 7). The propensity for ferroptosis appeared even more pronounced in A375AR cells, where HO-1 expression increased, while GPX4, SLC7A11, and FTH1 were significantly downregulated (Figure 7).

The FO-1AR cell population differs from A375AR in terms of total ERK expression and, consequently, in the expression of cell cycle-activating proteins, the expression levels of which remain elevated (Figure 5 and Figure 6). Furthermore, FO-1AR cells showed a lower propensity for ferroptosis but exhibited higher levels of apoptosis compared to A375AR (Figure 7).

### 2.5. Doravirine and Cabotegravir Inhibit Cell Cycle Progression and Induce Apoptosis in RAFi-Resistant Cell Populations

We investigated the molecular effects of antiretroviral drugs on RAFi-resistant cell populations, focusing specifically on apoptosis and cell cycle regulation.

After 72 h of treatment with doravirine (20 µM) in A375R and FO-1R cells, we observed an increase in cPARP expression levels. In contrast, cabotegravir (12 µM) elevated cPARP levels only in the FO-1R cell population (Figure 8).

P27^Kip1^ and P16^Ink4a^, both negative regulators of cyclin-dependent kinases (CDKs), inhibit cell proliferation when highly expressed. Immunoblot analysis revealed that P27^Kip1^ levels increased in FO-1R and A375AR cells, while P16^Ink4a^ expression was elevated only in A375DR and FO-1DR cells (Figure 8).

Our results demonstrate that both doravirine and cabotegravir can regulate the cell cycle and enhance apoptosis in RAFi-resistant melanoma cells.

### 2.6. RAF-Inhibitors and Antiretroviral Drugs Affect the Ability of Parental and Resistant Cell Lines to Form Colonies in Soft Agar

We aimed to evaluate the ability of RAF inhibitors and antiretroviral drugs to inhibit anchorage-independent cell growth in both parental and resistant cell populations using a soft agar colony formation assay.

Dabrafenib and AZ628, administered at 50 nM, strongly inhibited the anchorage-independent growth of the A375P and FO-1P cell lines, as expected. Interestingly, treatments with both RAF inhibitors of the A375DR, FO-1DR and FO-1AR cell populations conserved a significative reduction in both the size and number of colonies (Figure 9).

Our previous findings [20] indicated that cabotegravir and doravirine were effective in reducing the formation of soft agar colonies in A375P and FO-1P cells. In this work, we investigated their effectiveness in RAF-inhibitor-resistant cell populations. The treatment with 20 μM doravirine or 12 μM cabotegravir effectively decreased the number and size of the colonies in the FO-1R cell lines and almost completely prevented colony formation in A375DR cells. In contrast, A375AR cells exhibited low and slow growth even in untreated conditions, making the reduction in colony number with doravirine not significative. Cabotegravir, however, appeared to have a significative impact on growth in an anchorage-independent manner; meanwhile, AZ628 seems to not affect the growth much at all (Figure 9).

## 3. Discussion

Melanoma is a malignant tumor with a global presence, known for its rapid growth, early metastasis, recurrence, and poor prognosis [26]. Despite significant advancements in melanoma treatment, many patients have developed resistance to current therapies, highlighting the need for more effective options [2]. Melanoma’s aggressiveness is largely attributed to significant intra-tumoral heterogeneity, which contributes to treatment resistance and a high potential for spread [27]. The tumor’s heterogeneity is driven by transcriptionally distinct melanoma cell phenotypes, which can reprogram to adapt to different stages of progression and treatment exposure [27].

BRAF mutations, which occur in approximately 40–60% of melanoma cases, lead to the constitutive activation of MAPK pathway driving uncontrolled cell division and contributing to melanoma initiation and progression [28,29]. In vitro cellular models are valuable tools for mimicking and studying resistance mechanisms. Understanding the molecular transformations that occur in cells developing resistance to RAF inhibitors can guide the development of therapies to overcome this resistance and improve clinical outcomes.

Firstly, we examined the baseline phenotype of two BRAF-mutated human melanoma cell lines (A375P and FO-1P) and assessed the molecular signature changes after 4–5 months of treatment with increasing concentrations of the second-generation BRAF inhibitor dabrafenib and of the third-generation pan-RAF inhibitor AZ628.

As expected, both A375P and FO-1P cell lines were initially sensitive to the cytostatic/cytotoxic effects of the targeted drugs at the concentrations of 10 and 50 nM (Figure 1). However, prolonged treatment of the parental cells with either drug applied selective pressure, allowing the cells to survive at significantly higher concentrations, up to 5 µM. This indicates that, over the course of 4–5 months, both cell lines exhibit plasticity and can adapt to new conditions, developing drug-resistant phenotypes. Vitality assays, measuring fluorescence emission following DAPI staining, revealed that dabrafenib- or AZ628-treatment of A375R and FO-1R cells did not significantly reduce cell viability compared to untreated resistant-control cells, further confirming the development of drug resistance (Figure 2).

Therefore, we investigated the molecular signatures of A375R and FO-1R cells in comparison to their respective parental populations. Additionally, since the two targeted drugs have different mechanisms of action, we can simultaneously examine whether cell adaptation to each RAFi leads to distinct phenotypes.

Immunoblot analysis revealed that cell-cycle-regulating proteins (pRb, pCDC25C, cyclin D1, and cyclin A2), as markers of cell proliferation, were significantly downregulated by both targeted drugs in A375P and FO-1P cells, as expected (Figure 6). While A375DR and FO-1DR fully restored their ability to proliferate in the presence and absence of dabrafenib, FO-1AR only partially restored this ability, and A375AR remained in a state where cell cycle progression was completely hindered (Figure 6).

To further investigate the mechanisms underlying RAF inhibitor resistance, we focused on the oncoproteins that drive melanoma cell growth. BRAF overexpression or alternative splicing may contribute to BRAFi resistance [30,31]. Reactivation of ERKs can also occur through enhanced RAF dimerization, a key mechanism of resistance to RAFi in melanoma [32], as well as the abnormal expression of other RAF isoforms, such as CRAF and ARAF [14,33].

Consequently, we measured the expression levels of key proteins, including BRAF1, CRAF, and ERKs in both parental and resistant cell populations (Figure 5). Both RAF inhibitors downregulated the MAPK signaling pathway in parental cell lines and affected the total amount of ERK proteins in addition to their activation (Figure 5).

Receptor tyrosine kinases (RTKs) can function as upstream activators of the MAPK/ERK signaling pathway, and their increased expression has previously been observed in BRAFi-resistant cell lines [34,35,36]. Abnormal expression of RTKs has been shown to contribute to melanoma development and progression [23]. AXL enhances melanoma aggressiveness by activating the AKT (protein kinase B), p38 kinase, and MAPK signaling pathways [37]. The hepatocyte growth factor/MET pathway also plays a key role in melanoma development, progression, and therapeutic resistance [38].

The expression of two tyrosine kinase receptors, AXL and c-Met, was lower in parental populations treated with RAF inhibitors (Figure 5), reducing tumor aggressiveness when the cells were still sensitive. Resistant cells displayed distinct phenotypes after prolonged treatment with the BRAFi dabrafenib or the pan-RAF inhibitor AZ628. In dabrafenib-resistant cells, RTKs, MAPKs, and cell-cycle-regulating factors are reactivated, exhibiting a very high proliferative phenotype driven by the elevated expression of oncoproteins involved in growth signaling pathways. Other studies have reported that cells resistant to BRAF inhibitors, such as vemurafenib or dabrafenib, undergo metabolic reprogramming and reactivate several kinase pathways [16]. These cells also exhibit cell cycle deregulation [15]. Our findings in dabrafenib-resistant cells (A375DR and FO-1DR) are consistent with these previous reports. Conversely, in AZ628-resistant cell populations, these pathways remained almost inactive (Figure 5, Figure 6 and Figure 10). A375AR melanoma cells, characterized by slow-cycling or growth arrest and a senescent-like appearance, represent a different tumor signature (Figure 10). Although this phenotype is less aggressive, it requires alternative strategies, distinct from those targeting highly proliferative tumor cells, which instead leave the slow-cycling cancer cells unaffected [39].

The dysregulated expression of oncoproteins in the growth signaling pathway, which affects cell cycle regulation, should be assessed through the diverse functionalities of the two drugs. Notably, a major issue with second-generation BRAF inhibitors, such as dabrafenib, is that they are ATP-competitive and specifically target BRAF-mutated monomers and homodimers. However, they can induce a ‘paradoxical effect’ in non-tumor cells of melanoma patients, activating the MAPK pathway, which leads to cell proliferation and an increased risk of secondary malignancies [7]. In contrast, as above described, third-generation inhibitors also target CRAF in both monomeric and heterodimeric forms through an allosteric, non-competitive mechanism [7]. Only dabrafenib-resistant cells showed re-expression of BRAF along with CRAF expression. These events facilitate the bypass of BRAF inhibition by promoting the formation of BRAF-CRAF heterodimers. Conversely, the pan-RAFi AZ628 permanently blocks both CRAF or BRAF-CRAF activity. Accordingly, some studies reported that AZ628 has a superior ability to block the MAPK signaling pathway compared to the second-generation BRAFi vemurafenib [40,41], as well as a high binding affinity to other intracellular targets involved in proliferation and migration pathways [42].

Considering these different factors, it is crucial to investigate the propensity to cell death across all cell populations. Cleaved PARP expression is a key indicator that the execution phase of apoptosis has begun [24]. After 72 h of treatment with either dabrafenib or AZ628, cPARP levels increased in all sensitive parental cell lines (Figure 7), highlighting the potent cytotoxic effects of RAF inhibitors. Unexpectedly, only A375AR cells completely suppressed apoptosis. Conversely, dabrafenib-resistant cells (A375DR and FO-1DR) and FO-1AR maintained high levels of cPARP, both in the presence and absence of the drug, suggesting that some cells continued to undergo apoptosis after acquiring resistance (Figure 7 and Figure 10). To explore the regulation of this process, we measured the expression of proteins involved in apoptotic cell death [43], including the antiapoptotic proteins Bcl2 and Bcl-xL, and the proapoptotic proteins Bax and MTCH2. It was found that in resistant melanoma cell lines, the expression of the proapoptotic proteins BAX and MTCH2 were negatively or not correlated to cPARP levels. Additionally, the same cell population exhibited a positive correlation between the expression of the antiapoptotic proteins Bcl2 or Bcl-xL and cPARP. Strikingly, studies in the literature have also reported similarly contradictory findings in various tumors, including melanoma [44,45,46]. However, our immunoblots suggest that additional proteins or mechanisms may be involved.

If the plasticity and dedifferentiation of melanoma cells promote resistance to targeted therapies, the newly acquired phenotype might also render the cells more susceptible to other vulnerabilities. Therefore, we investigated whether melanoma cells can undergo ferroptosis, a non-apoptotic form of cell death triggered by iron. Ferroptosis is induced by an increase in intracellular free iron, which can occur due to high transferrin uptake, heme oxygenase 1 (HO-1) induction, and/or low expression of ferritin (FTH1), a protein that sequesters iron in its cage [47,48]. Free iron can induce lipid peroxidation and cell death, but only if its damaging effects are not counteracted by specific intracellular antioxidant systems, such as glutathione peroxidase 4 (GPX4) activity and the availability of glutathione, which is highly dependent on the cystine transporter SLC7A11 [49,50].

In general, our immunoblot data suggest that A375 cell line is more susceptible to ferroptosis than FO-1 cells (Figure 7 and Figure 10). Specifically, A375P cells treated with both RAF inhibitors and FO-1P cells treated with dabrafenib exhibited protein expression signatures indicative of ferroptosis, including high levels of transferrin and HO-1, along with low expression of SLC7A11 and GPX4/FTH1 (Figure 7). In contrast, FO-1DR cells appeared to be the most resistant population to ferroptosis, while A375DR cells showed only weak signs of susceptibility to this type of cell death. However, AZ628-resistant cells, particularly A375AR, seemed prone to ferroptosis (Figure 7 and Figure 10). Notably, the A375AR population displayed a low proliferative and low apoptotic phenotype but showed a strong propensity to ferroptosis, with high expression of transferrin and HO-1, and very low expression of FTH1, GPX4, and SLC7A11.

In summary, the differences in RAF-inhibitor-resistant cells are dependent on the type of RAFi used to induce resistance. Dabrafenib-resistant cells are highly proliferative, with a high propensity for apoptosis, but not for ferroptosis, and can be hindered by blocking cell cycle progression (Figure 10). In contrast, AZ628-resistant cells, particularly the A375AR population, exhibit a low proliferative rate and low apoptosis but have a high propensity for ferroptosis (Figure 10). These cells can be targeted with inducers of either apoptosis or ferroptosis.

We and others have previously demonstrated that some antiretroviral drugs, typically used to combat retroviruses, with a particular focus on HIV infection, can reduce melanoma aggressiveness by inhibiting the expression of human endogenous retrovirus (HERV-K) [20,51]. Moreover, several studies have reported the antitumor effects of antiretroviral drugs in other malignancies as well [52,53,54,55,56,57,58]. In this work, we confirmed the ability of doravirine and cabotegravir to reduce cell viability in parental cells, as previously reported [20], and we analyzed their activity on resistant cell populations. While cabotegravir appears equally effective against both A375P and FO-1P cells, doravirine shows greater efficacy in A375P compared to FO-1P, consistent with our previous findings [20] (Figure 1). In cell populations that have acquired resistance to RAFi, both doravirine and cabotegravir remain effective, particularly in A375DR cells, where they reduced cell viability more than in A375P cells (Figure 2). Furthermore, both antiretroviral drugs significantly reduced the number of colonies in soft agar, suggesting that they retain the ability to inhibit cell growth in an anchorage-independent manner (Figure 9).

According to the literature, RAFi resistance is associated with a more aggressive tumor phenotype [59]. Moreover, as highlighted by Serafino et al. [60], the activation of human endogenous retroviral genes may contribute to the worsening of the melanoma phenotype in patients. Our real-time-PCR data showed that both dabrafenib and AZ628 reactivated the expression of *HERV-K Pol* and *Env* genes compared to untreated parental cells (Figure 3). These results highlight the need for further investigation into HERV-K gene expression levels in resistant cell populations.

A general reduction in HERV-K gene expression was observed in all four resistant cell lines following treatment with either antiretroviral drug (Figure 4). This suggests that the decrease in HERV-K gene expression may contribute to the antimelanoma effects of antiretroviral drugs, even in the context of RAF inhibitor resistance, which reactivates HERV-K *Pol* and *Env* expression.

At the molecular level, antiretroviral drugs may influence the expression of key regulatory proteins involved in the cell cycle and cell death. In our previous study, cabotegravir demonstrated pro-apoptotic activity in FO-1P cells after 48 h of treatment [20]. In FO-1 cells resistant to RAFi, cabotegravir induced cPARP expression after 72 h of treatment, confirming its cytotoxic effects in these resistant cell lines as well (Figure 8). Although doravirine did not induce apoptotic cell death in A375P and FO-1P cells after 48 h of treatment [20], it led to an increase in cPARP expression across all RAFi-resistant cell populations after 72 h (Figure 8). Notably, in the same cell populations, doravirine showed stronger inhibition of HERV-K *Pol* and *Env* expression compared to cabotegravir (Figure 4). Additionally, doravirine effectively reactivated apoptosis in the A375AR cell population, where long-term AZ628 treatment had previously suppressed apoptotic activity (Figure 8).

A substantial reduction in the expression of p16^Ink4a^ and p27^Kip1^ has been observed with melanoma progression [61]. These proteins act as tumor suppressors by inhibiting CDKs, which play a crucial role in regulating cell cycle checkpoint transitions [62,63]. Doravirine significantly increased p16^Ink4a^ expression in dabrafenib-resistant cells (A375DR and FO-1DR), indicating a potential reduction in cell growth within these highly proliferative resistant populations (Figure 8). Additionally, both doravirine and cabotegravir elevated P27^Kip1^ expression in FO-1DR cells as well as in AZ628-resistant cells (A375AR and FO-1AR).

In summary, antiretroviral drugs may influence both apoptosis and cell proliferation in RAFi-resistant melanoma cell lines, partly by targeting cell cycle regulation and promoting apoptotic cell death.

## 4. Materials and Methods

### 4.1. Cell Cultures

The A375 (CRL-1619) and FO-1 (CRL-12177) melanoma cell lines were obtained from ATCC (Manassas, VA, USA) and cultured at 37 °C in a humidified atmosphere containing 5% CO_2_. The culture medium used was high-glucose Dulbecco’s Modified Eagle’s Medium (DMEM, Gibco, BRL Invitrogen Corp., Carlsbad, CA, USA), supplemented with 10% heat-inactivated fetal bovine serum (FBS, Euroclone S.p.A, Pero, Milan, Italy) and 1% antibiotic–antimycotic solution (Gibco, BRL Invitrogen Corp., Carlsbad, CA, USA).

### 4.2. Cell Viability Assay (DAPI Staining and Measure of Fluorescence)

4′,6-diamidino-2-phenylindole dihydrochloride (DAPI) is a fluorescent compound that exhibits multiple binding modes to DNA [22,64] and is transferred to descendant cells during proliferation.

Melanoma cells were plated in 96-well black plates with clear bottoms (LUMOX multiwell, Sarstedt AG & Co. KG, Nümbrecht, Germany). The following cell lines were used: A375P, A375DR, and A375AR melanoma cells (1.45 × 10^3^ cells/well); and FO-1P, FO-1DR, and FO-1AR melanoma cells (1.75 × 10^3^ cells/well).

The next day, the cells were treated with antiretroviral drugs or the kinase inhibitors dabrafenib (GSK2118436, Selleckchem, Huston, TX, USA) or AZ628 (Merck, Milan, Italy) for 72 h. After the treatment period, the medium was removed by vacuum, and the cells were fixed by adding 50 μL/well of 4% (*w*/*v*) paraformaldehyde (PFA) (AppliChem, Monza, Italy). After 10 min of incubation, the PFA was removed by vacuum, and the wells were washed three times for 10 min each with 1× Dulbecco’s Phosphate-Buffered Saline (DPBS) (Gibco, BRL Invitrogen Corp., Carlsbad, CA, USA) on a shaker.

Subsequently, 50 μL/well of DAPI (1:1000 dilution, Thermofisher Scientific, Milan, Italy) was added, and the plates were incubated at room temperature (RT) for 15 min in the dark on a shaker. The DAPI was then removed by vacuum, and the wells were washed three times for 10 min each with 1× DPBS, again in the dark.

Finally, fluorescence intensity was measured at an excitation wavelength of 350 nm and an emission wavelength of 461 nm using a TECAN NanoQuant Infinite M200 Pro plate reader (Tecan Group Ltd., Männedorf, Switzerland). Each experimental condition was performed in 8 replicates.

### 4.3. RNA Extraction, Reverse Transcription, and Real-Time PCR

A375P, A375R, FO-1P, and FO-1R cell lines were seeded in 6 mm petri dishes at the following densities: A375P and FO-1P 100 × 10^3^ cells/dish; A375DR, FO-1DR, and FO-1AR 150 × 10^3^ cells/dish; A375AR 180 × 10^3^ cells/dish. Upon reaching 80% confluence, all cell lines were treated with either 20 µM doravirine, 12 µM cabotegravir (Merck, Milan, Italy), 50 nM dabrafenib (GSK2118436, Selleckchem, Huston, TX, USA) or 50 nM AZ628 (Merck, Milan, Italy). After 24 h of treatment, total RNA was extracted for gene expression analysis using TRIzol Reagent (Thermofisher Scientific, Milan, Italy), following the manufacturer’s protocol. RNA quantification was performed using a Tecan NanoQuant Infinite M200 Pro plate reader (Tecan Group Ltd., Männedorf, Switzerland), and RNA quality was assessed via 1% agarose gel electrophoresis.

A total of 1000 ng of RNA was reverse transcribed using the SensiFAST cDNA Synthesis Kit (Bioline, Trento, Italy) in accordance with the manufacturer’s instructions. Following reverse transcription, the expression levels of HERV-K genes (*Pol, Env*) were measured using Real-Time Polymerase Chain Reaction (RT-PCR). Normalization was carried out using Cluster of Differentiation 151 protein (*CD151*), which was identified as the most stable gene under our experimental conditions.

The primers used for amplification were as follows:

*CD151*: (Fw) 5′-CTACGCCTACTACCAGCAGC-3′

(Rv) 5′-CGGAACCACTCACTGTCTCG-3′

*Pol*: (Fw) 5′-CCACTGTAGAGCCTCCTAAACCC-3′

(Rv) 5′-GCTGGTATAGTAAAGGCAAATTTTTC-3′

*Env*: (Fw) 5′-GCCATCCACCAAGAAAGCA-3′

(Rv) 5′-AACTGCGTCAGCTCTTTAGTTGT-3′

Real-Time PCR was performed using the Bio-Rad CFX Connect Real-Time System and the SensiFAST SYBR No-ROX Kit (Bioline, Trento, Italy). The amplification protocol consisted of an initial polymerase activation step at 95 °C for 2 min, followed by 40 cycles of denaturation at 95 °C for 5 s and primer annealing/polymerization at 60 °C for 20 s. Each measurement was performed in triplicate in at least four independent experiments.

### 4.4. Colony Formation Assay in Soft Agar

The ability of FO-1P, A375P, A375R, and FO-1R melanoma cells to grow in an anchorage-independent manner was evaluated using a colony formation assay in soft agar, as described previously [65]. First, the bottom layer of 6-well plates was prepared with 1% low-gelling temperature agarose (Sigma-Merck, Milan, Italy) dissolved in 2× DMEM, 20% FBS, and 2% antibiotic–antimycotic solution. A second layer, consisting of 0.6% low-gelling temperature agarose dissolved in the same medium, was added on top. This layer contained treated or untreated melanoma cells (A375P, and FO-1P 12 × 10^3^ cells/well; A375R 20 × 10^3^ cells/well; FO-1R 10 × 10^3^ cells/well).

Fresh medium (200 μL) was added to each well twice a week. After 15–21 days, colony formation was assessed using an inverted microscope (Axio Vert A1, Zeiss, Oberkochen, Germany).

### 4.5. Total Protein Extraction

Cells were seeded in 35 mm petri dishes at the following densities: A375P 40 × 10^3^ cells/dish; FO-1P 50 × 10^3^ cells/dish; A375DR and FO-1R 60 × 10^3^ cells/dish; A375AR 80 × 10^3^ cells/dish. After 24 h, cells were treated with each drug or left untreated as a control. After 72 h of treatment, the cells were scraped using warm 1× sample buffer (2% SDS, 10% glycerol, 50 mM Tris-HCl, 1.75% β-mercaptoethanol, and bromophenol blue), boiled at 98 °C and then put in ice for 10 min. The total protein extracts were stored at −80 °C for subsequent analysis.

### 4.6. Immunoblot Analysis

Protein extracts were separated by electrophoresis on 10 to 15% polyacrylamide SDS-PAGE gels and then transferred onto polyvinylidene difluoride (PVDF) membranes (Merck-Millipore, Milan, Italy). Membranes were blocked at room temperature (RT) for 1 h in TBST buffer (10 mM Tris-HCl, pH 7.5, 100 mM NaCl, 0.1% Tween 20) containing 5% milk. Following blocking, the membranes were incubated overnight at 4 °C on a shaker in a 5% BSA solution containing the following primary antibodies: Transferrin (A1448), Ferritin heavy chain (A19544), cleaved PARP-P25 (A19612), CDKN1B/p27KIP1 (A19095), CDKN2A/p16INK4a (A11651), Cyclin A2 (A19036), Bcl-xL (A0209), Heme Oxygenase 1 (A19062), ERK 1/2 (A16686), pERK 1/2 (AP0974), SLC7A11/xCT (A2413), C-Raf (A19638) from ABclonal (Woburn, MA, USA); Cyclin D1 (GTX634347), MTCH2 (GTX130324), β-actin (GTX124214), p21 Cip (GTX29543), CDC25C phospho Ser216 (GTX128153), BAX (GTX109683), c-Met (GTX100637), B-Raf1 (GTX100913) from Genetex (Alton Parkway, Irvine, CA, USA); Bcl-2 (#15071), pRb (#8516) from Cell Signaling Technology (Danvers, MA, USA) and AXL (13196-1-AP), GPX4 (67763-1) from ProteinTech (Manchester, UK).

After primary antibody incubation, the membranes were washed three times for 10 min each with TBST buffer, followed by a 1 h incubation with horseradish peroxidase-conjugated secondary antibodies (anti-rabbit or anti-mouse) from Cell Signaling Technology (Danvers, MA, USA). The membranes were then washed again three times for 10 min each with TBST. Protein expression levels were normalized to β-actin unless otherwise specified. Immunodetection was performed using an ECL detection kit (Cyanagen, Bologna, Italy), and chemiluminescence signals were visualized using a ChemiDoc system (Bio-Rad, Hercules, CA, USA).

### 4.7. Statistical Analysis

Data are presented as mean ± standard deviation (S.D.). Statistical differences were analyzed using GraphPad Prism software, version 8.0.2, employing an unpaired two-tailed Student’s *t*-test, unless otherwise specified. A *p*-value of less than 0.05 (*) or less than 0.01 (**) was considered statistically significant. Each experiment was performed with a minimum of three independent biological replicates. Normal data distribution was verified using the Shapiro–Wilk test.

## 5. Conclusions

RAF inhibitor therapy creates resistance in vitro and in vivo. Targeting other molecular pathways may provide therapeutic benefits when RAF inhibitors lose effectiveness. Our work demonstrates that antiretroviral drugs, doravirine and cabotegravir, not only induce apoptosis but also modulate the expression of cell cycle regulators in RAFi-resistant melanoma cells. These antiretrovirals, which have been used long-term with minimal side effects for AIDS treatment, show promise in inhibiting tumor growth in RAFi-resistant cell lines, suggesting their potential utility in slowing melanoma progression after RAFi resistance. We assume that these findings should be validated in preclinical studies.

## Figures and Tables

**Figure 1 ijms-25-11939-f001:**
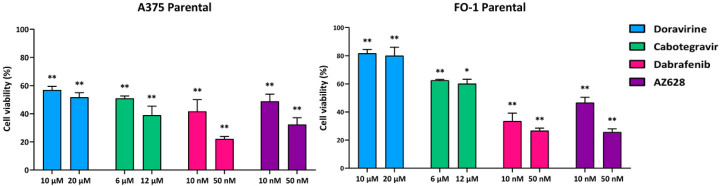
Parental A375 and FO-1 cell viability after antiretroviral or RAF inhibitor treatments. Cell viability was assessed using the DAPI fluorometric assay after 72 h of treatment with doravirine, cabotegravir, dabrafenib, or AZ628 in A375 and FO-1 cell lines. Data were acquired by calculating the mean ± S.D. of values from four independent experiments, each conducted in eight technical replicates, and then compared with the untreated control. Statistical significance was indicated as * *p* < 0.05; ** *p* < 0.01.

**Figure 2 ijms-25-11939-f002:**
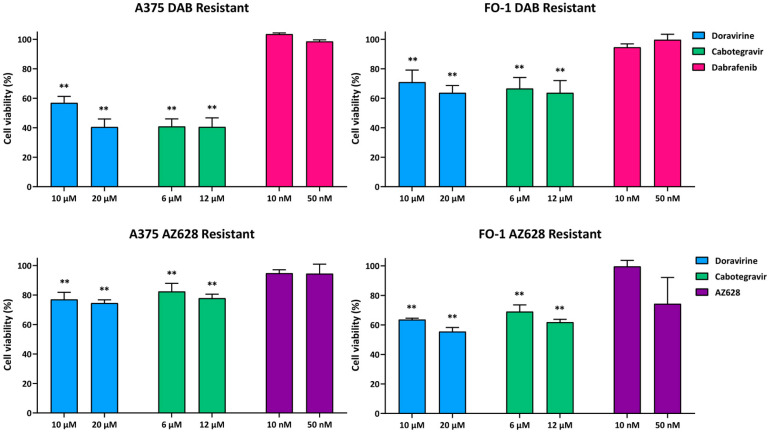
Cell viability of dabrafenib- or AZ628-resistant cell populations after antiretroviral treatments. Cell viability was assessed using the DAPI fluorometric assay after 72 h of treatment with either doravirine, cabotegravir, dabrafenib or AZ628 in A375R, and FO-1R cell lines. Data were acquired by calculating the mean ± S.D. of values from four independent experiments, each conducted in eight technical replicates, and then compared with the untreated control. Statistical significance was indicated as ** *p* < 0.01.

**Figure 3 ijms-25-11939-f003:**
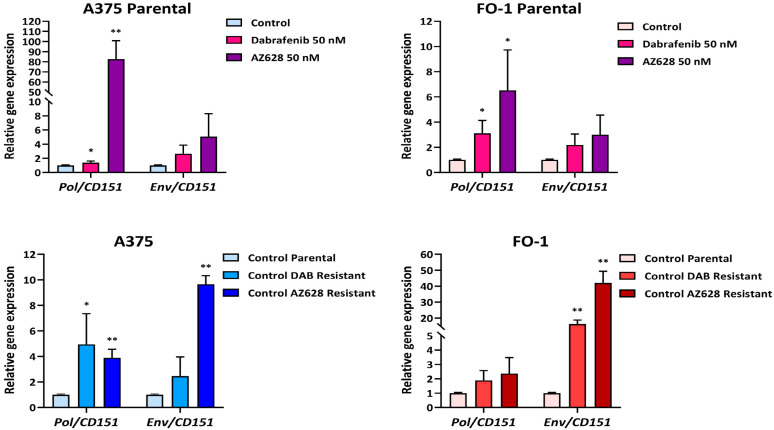
Relative expression of HERV-K *Pol* and *Env* genes in parental and RAFi-resistant cell populations after treatment with RAF inhibitors. Cells were treated with dabrafenib (50 nM) or AZ628 (50 nM) for 24 h. The mRNA expression levels of treated samples were measured by RT-PCR and compared with untreated cells of each population. The bars represent the mean values ± standard deviation (S.D.) of four independent experiments, each conducted in triplicates. All comparisons were made against each control sample after normalization with CD151 expression. Statistical significance was indicated as * *p* < 0.05; ** *p* < 0.01.

**Figure 4 ijms-25-11939-f004:**
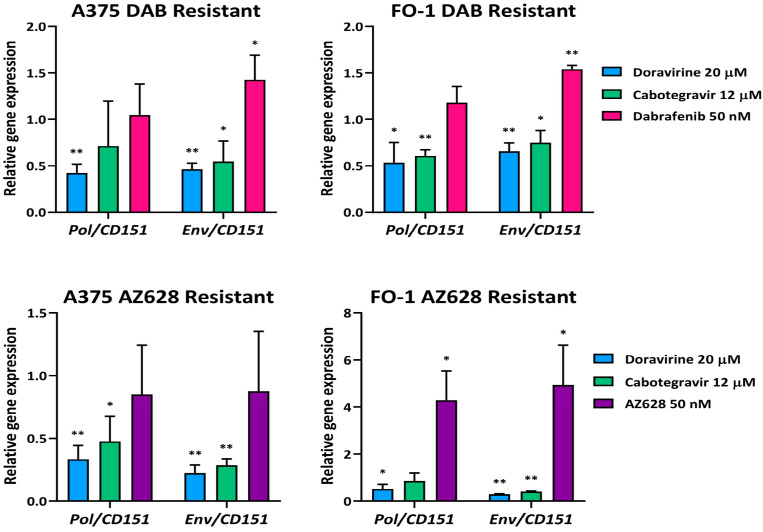
Relative expression of HERV-K *Pol* and *Env* genes in different RAFi-resistant cell populations. Cells were treated with doravirine (20 µM), cabotegravir (12 µM), dabrafenib (50 nM) or AZ628 (50 nM) for 24 h. The mRNA expression levels of treated samples were measured by RT-PCR and compared with untreated cells of each population. The bars represent the mean values ± standard deviation (S.D.) of four independent experiments, each conducted in triplicates. All comparisons were made against each control sample after normalization with CD151 expression. Statistical significance was indicated as * *p* < 0.05; ** *p* < 0.01.

**Figure 5 ijms-25-11939-f005:**
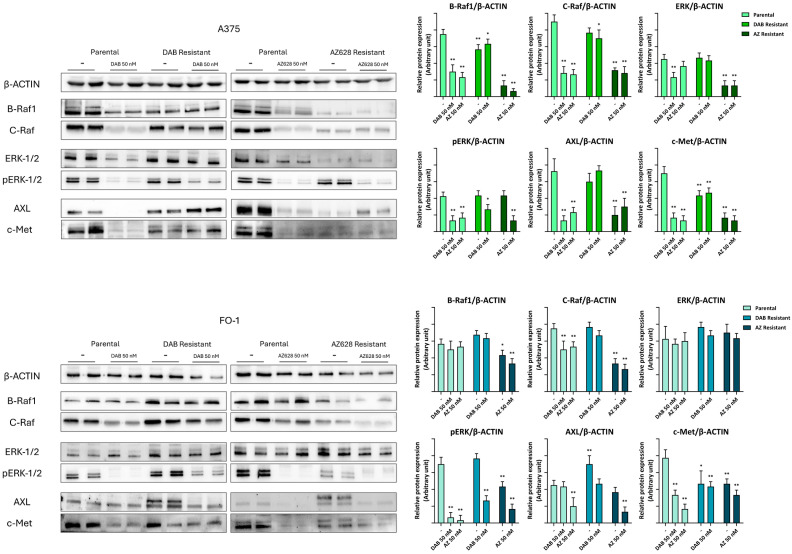
Representative immunoblots show the effects of RAF inhibitors on the expression of oncoproteins regulating MAPK pathway. Dabrafenib (50 nM) or AZ628 (50 nM) treatment regulate B-Raf1(B-V-raf murine sarcoma viral oncogene), C-Raf (C-V-raf murine sarcoma viral oncogene), ERK (extracellular signal-regulated kinases), pERK, AXL (tyrosine-protein kinase receptor UFO), and c-Met (mesenchymal–epithelial transition factor) protein expressions in both parental and resistant cell populations. The histograms represent the mean values ± S.D. of protein expression levels measured by densitometry deriving from three independent experiments and normalized to β-actin expression. All comparisons were made relative to each control sample after data normalization. Statistical significance was indicated as * *p* < 0.05; ** *p* < 0.01.

**Figure 6 ijms-25-11939-f006:**
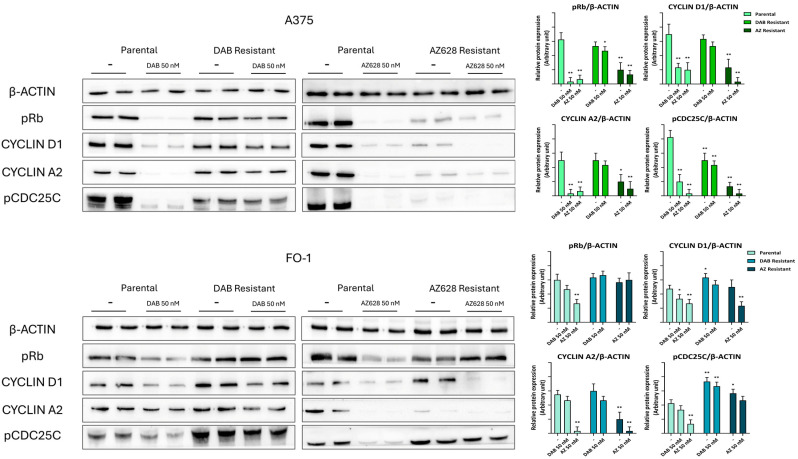
Representative immunoblots show the effects of RAF inhibitors on the expression of proteins regulating cell cycle. Dabrafenib (50 nM) or AZ628 (50 nM) treatment downregulates cell cycle regulatory proteins in both parental and resistant cell populations. pRb (phosphorylated form of retinoblastoma protein), pCDC25C (phosphorylated form of M-phase inducer phosphatase 3). The histograms represent the mean values ± S.D. of protein expression levels measured by densitometry deriving from three independent experiments and normalized to β-actin expression. All comparisons were made relative to each control sample after data normalization. Statistical significance was indicated as * *p* < 0.05; ** *p* < 0.01.

**Figure 7 ijms-25-11939-f007:**
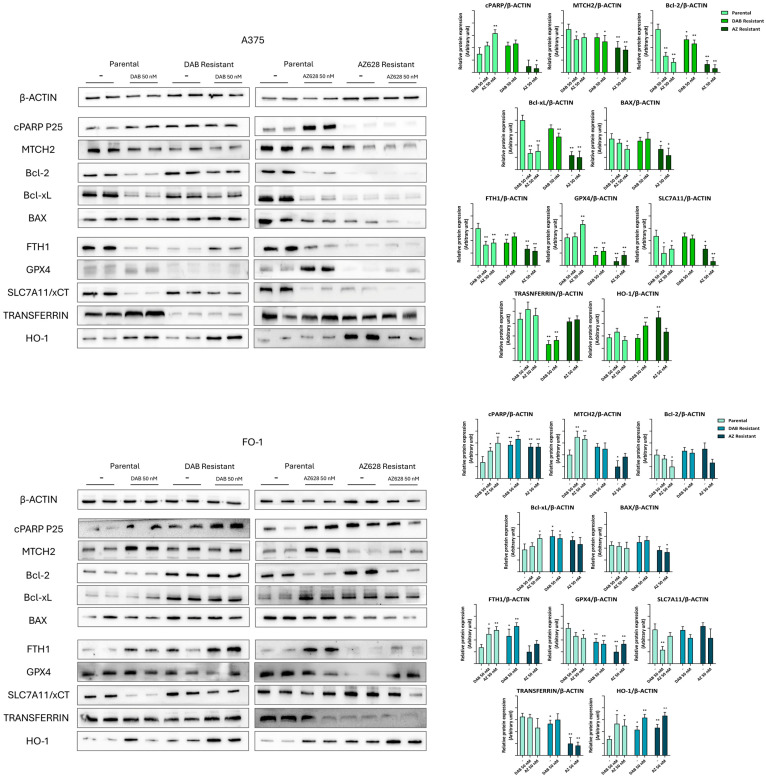
Representative immunoblots show the effects of RAF inhibitors on the expression of proteins regulating cell death in parental and resistant cell populations. Dabrafenib induced apoptosis in the parental melanoma cells as well as in A375DR, FO-1DR and FO-1AR, as suggested by the high expression of the cleaved form of poly (ADP-ribose) polymerase (cPARP). Ferroptosis is induced in all parental cells lines and in A375AR cell population, as suggested by a decrease in ferritin (FTH1), cystine-glutamate antiporter (SLC7A11) and/or glutathione peroxidase 4 (GPX4) expression and increased transferrin or heme oxygenase-1 (HO-1). The histograms represent the mean values ± S.D. of protein expression levels measured by densitometry deriving from three independent experiments and normalized to β-actin expression. All comparisons were made relative to each control sample after data normalization. Statistical significance was indicated as * *p* < 0.05; ** *p* < 0.01.

**Figure 8 ijms-25-11939-f008:**
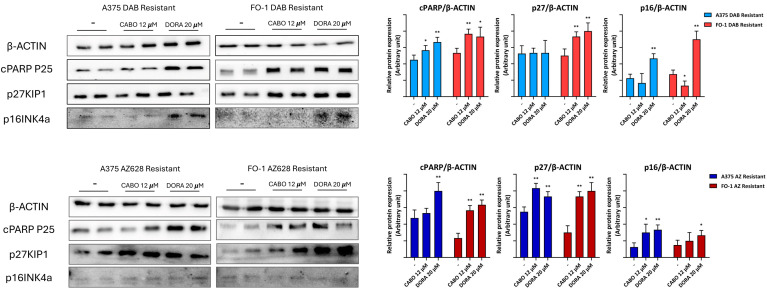
Representative immunoblots show the effects of antiretroviral drugs on the expression of proteins regulating cell cycle and apoptosis in resistant cell populations. Cabotegravir (CABO) and doravirine (DORA) induced apoptosis in FO-1R cells, while doravirine also induced apoptosis in A375R cell populations, as indicated by the high expression of cPARP. The expression of tumor suppressor protein p27 increased in FO-1R and in A375AR cell populations when treated with either cabotegravir or doravirine. The expression of tumor suppressor protein p16 increased only in dabrafenib-resistant cell populations. The histograms represent the mean values ± S.D. of protein expression levels measured by densitometry deriving from three independent experiments and normalized to β-actin expression. All comparisons were made relative to each control sample after data normalization. Statistical significance was indicated as * *p* < 0.05; ** *p* < 0.01.

**Figure 9 ijms-25-11939-f009:**
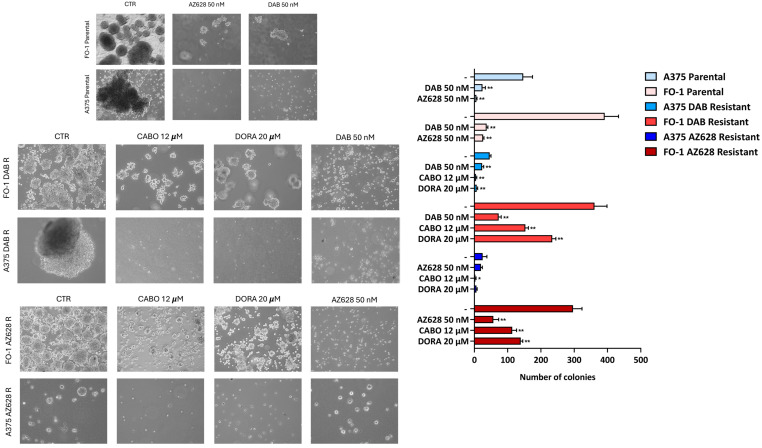
Effects of RAF inhibitors and antiretroviral drugs in soft agar colony formation in melanoma parental and resistant cell populations. The images shown are representative examples from experiments illustrating colony formation in soft agar. The histograms show the quantification of colony numbers for each treatment, based on data from three independent experiments, each performed in duplicate. Microscopic images depict the size and density of colonies (5× magnification, inverted microscopy, Axio Vert A1, Zeiss, Oberkochen, Germany). All comparisons were made against each respective control sample. Statistical significance was indicated as * *p* < 0.05; ** *p* < 0.01.

**Figure 10 ijms-25-11939-f010:**
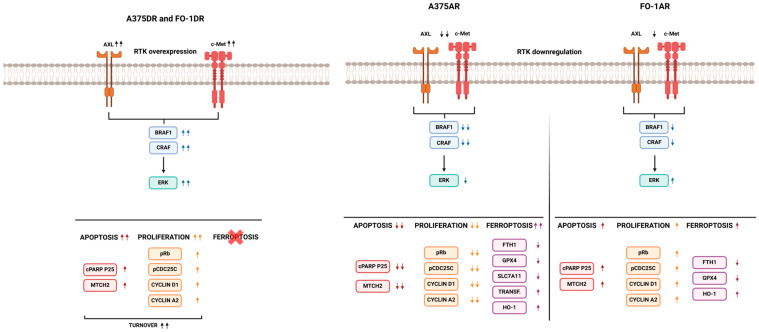
Characterization of phenotypes in melanoma cell populations resistant to dabrafenib or AZ628. This figure illustrates various features of melanoma cell populations with respect to signaling pathways, cell cycle regulation, and susceptibility to cell death. Receptor tyrosine kinase (RTK): including tyrosine-protein kinase receptor UFO (AXL) and mesenchymal–epithelial transition factor (c-Met). RAS/RAF/MAPK signaling pathway: detailing expression levels of V-raf murine sarcoma viral oncogenes (RAF) including BRAF and CRAF isoforms, as well as total extracellular signal-regulated kinases (ERK) expression. Pro-proliferative proteins: including phosphorylation levels of retinoblastoma protein (pRb) and M-phase inducer phosphatase 3 (pCDC25C), along with cyclin D1 and cyclin A2 expression. Apoptosis markers: displaying levels of cleaved poly (ADP-ribose) polymerase (cPARP) and mitochondrial carrier homolog 2 (MTCH2). Ferroptosis markers: highlighting the expression of transferrin and heme oxygenase-1 (HO-1), both of which contribute to increased intracellular iron. Levels of ferritin heavy chain 1 (FTH1), glutathione peroxidase 4 (GPX4), and cystine-glutamate antiporter (SLC7A11) are also presented.

## Data Availability

Data contained within the article are recorded using instruments in our labs. These instruments are not connected to the internet; however, further inquiries can be directed to the corresponding author.

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
