# Peer review of "BRAF-Mutated Melanoma Cell Lines Develop Distinct Molecular Signatures After Prolonged Exposure to AZ628 or Dabrafenib: Potential Benefits of the Antiretroviral Treatments Cabotegravir or Doravirine on BRAF-Inhibitor-Resistant Cells"

_ijms, 2024, doi:10.3390/ijms252211939_

Round 1
Reviewer 1 Report
Comments and Suggestions for Authors
Dear Authors,
I believe that the manuscript is prepared very well and should be accepted in IJMS after minor revision.
In a revised version please include the number of technical and biological repeats of each experiment in figure captions.
Additional comments:
The submission by Zanrè et al. extends their previous work on BRAF-mutated melanoma cell resistance, where they examined the long-term effects of RAF inhibitors on melanoma cell lines. In this study, the authors focus on the development of distinct molecular signatures in melanoma cells resistant to RAF inhibitors AZ628 or dabrafenib. They explore how antiretroviral treatments, doravirine and cabotegravir, affect cell viability and apoptosis in these resistant cells. The results suggest that both antiretroviral drugs can potentially reverse drug resistance by inducing apoptosis and reducing cell viability. These findings open up new therapeutic avenues for overcoming RAF-inhibitor resistance in melanoma treatment.
I have some minor points that needs to be addressed in a revised version:
- the number of both technical and biological repeats should be noted in the captions below figures
- fig. 2: The authors use DAPI staining for cell viability. Would it be useful to include an additional control for debris levels to ensure the observed changes in fluorescence are not influenced by debris?
- I suggest that the authors also cite and discuss the following relevant publication, which explores resistance mechanisms in BRAF-mutated melanoma: Datta et al. (2024), Exp Cell Res, 442, 114215, doi:10.1016/j.yexcr.2024.114215.
Author Response
Dear Autors,
I believe that the manuscript is prepared very well and should be accepted in IJMS after minor revision.
In a revised version please include the number of technical and biological repeats of each experiment in figure captions.
Additional comments:
The submission by Zanrè et al. extends their previous work on BRAF-mutated melanoma cell resistance, where they examined the long-term effects of RAF inhibitors on melanoma cell lines. In this study, the authors focus on the development of distinct molecular signatures in melanoma cells resistant to RAF inhibitors AZ628 or dabrafenib. They explore how antiretroviral treatments, doravirine and cabotegravir, affect cell viability and apoptosis in these resistant cells. The results suggest that both antiretroviral drugs can potentially reverse drug resistance by inducing apoptosis and reducing cell viability. These findings open up new therapeutic avenues for overcoming RAF-inhibitor resistance in melanoma treatment.
Answer: We are very pleased that our work has been appreciated. We thank the reviewer for the valuable suggestions provided to improve this manuscript.
I have some minor points that need to be addressed in a revised version:
- the number of both technical and biological repeats should be noted in the captions below figures
Answer: Thank you for the suggestion. All requested information has been added to the figure captions.
- fig. 2: The authors use DAPI staining for cell viability. Would it be useful to include an additional control for debris levels to ensure the observed changes in fluorescence are not influenced by debris?
Answer: To explain the choice of DAPI for measuring cell viability, it’s important to note that this decision was based on findings from previous experiments. In our earlier studies, we used sulforhodamine B (SRB), a colorimetric assay that binds to the total protein mass in cells. While SRB is highly reproducible, we found that its results were understated compared to direct cell counts. This discrepancy might be due to the step where the entire cell culture medium is precipitated with trichloroacetic acid, which includes cell debris in the final measurement. DAPI, however, binds specifically to double-stranded DNA and is therefore not affected by cellular debris. Additionally, as described in the Materials and Methods section, we wash the wells three times with DPBS before and after DAPI staining, which helps prevent any nonspecific interactions.
- I suggest that the authors also cite and discuss the following relevant publication, which explores resistance mechanisms in BRAF-mutated melanoma: Datta et al. (2024), Exp Cell Res, 442, 114215, doi:10.1016/j.yexcr.2024.114215.
Answer: Thank you for the suggestion. We have already cited the paper by Datta et al. in the Introduction section. In the revised version of manuscript, we have also added the following sentences to the Discussion section (lines 430-434). 'Other studies have reported that cells resistant to BRAF inhibitors, such as vemurafenib or dabrafenib, undergo metabolic reprogramming and reactivate several kinase pathways (Datta et al.). These cells also exhibit cell cycle deregulation (Dulgar et al.). Our findings in dabrafenib-resistant cells (A375DR and FO-1DR) are consistent with these previous reports (Figure 10).
Reviewer 2 Report
Comments and Suggestions for Authors
Dear authors,
Congratulation on you excellent work! It was a pleasure reading your well documented and skillfully written manuscript.
However, I have the following suggestions:
1. Please consider including more precise data on response rates, overall survival and resistance rates for BRAF inhibitors.
2. I believe the manuscript would benefit from a diagram illustrating RAS/MAPK signaling pathway.
Best regards!
Author Response
Dear authors,
Congratulation on you excellent work! It was a pleasure reading your well documented and skillfully written manuscript.
Answer: We are very pleased that our work has been appreciated. We thank the reviewer for the valuable suggestions provided to improve this manuscript.
However, I have the following suggestions:
- Please consider including more precise data on response rates, overall survival and resistance rates for BRAF inhibitors.
Answer: We agree with the reviewer’s suggestion. In place of the more concise description in the previous version, we have now expanded subsection 2.2 to include all mean cell viability measurements along with their respective standard deviations. We added lines: 117-122; 137-141; 145-150; 152-163.
- I believe the manuscript would benefit from a diagram illustrating RAS/MAPK signaling pathway.
Answer: Thank you for the suggestion. We agree that a schematic figure can convey a more impactful message. However, rather than including a representative image of the widely known MAPK signaling pathway, we chose to visually highlight the distinct characteristics of dabrafenib-resistant cells (A375DR and FO-1DR) compared to AZ628-resistant cells (Figure 10). Notably, A375AR cells show more subtle differences compared to FO-1AR cells; thus, they are represented in a distinct way.